

# Stimulus–response complexity influences task-set inhibition in task switching

Li Zhao[1,2,3,*], Saisai Hu[1,2,3,*], Yingying Xia[1,2,3], Jinyu Li[1,2,3], Jingjing Zhao[1,2,3], Ya Li[1,2,3] and Yonghui Wang[1,2,3]

[1] School of Psychology, Shaanxi Normal University, Xi'an, China
[2] Shaanxi Provincial Key Laboratory of Behavior & Cognitive Neuroscience, Xi'an, China
[3] Shaanxi Provincial Key Research Center of Child Mental and Behavioral Health, Xi'an, China
* These authors contributed equally to this work.

## ABSTRACT

Previous studies have found that inhibiting a task set plays an important role in task switching. However, the impact of stimulus–response (S–R) complexity on this inhibition processing has not been explored. In this study, we applied the backward inhibition paradigm (switching between tasks A, B, and C, presented in sets of three) in order to investigate inhibition performance under different S–R complexities caused by corresponding S–R mappings. The results showed that the difficult condition resulted in a greater switch cost than the moderate and easy conditions. Furthermore, we found a significant $n-2$ repetition cost under the easy S–R complexity that was reversed under the difficult S–R complexity. To verify stability of the reversed $n-2$ repetition cost in the difficult condition, we recruited another independent sample to conduct an additional experiment with the difficult condition. These results replicated the reversed $n-2$ repetition cost. These findings suggest that S–R complexity affects task-set inhibition in task switching because the effect of the task-set inhibition was insignificant when the S–R complexity increased; it was only significant under the easy condition. This result was caused by the different cognitive resource assignments.

## INTRODUCTION

People must be highly adaptable and flexible when faced with complex environmental changes. This often involves engaging in complex cognitive control processes in order to more actively perform tasks and achieve goals. Task switching is a concept that has been well-established in prior studies on cognitive control (*Monsell, 2003*; *Kiesel et al., 2010*; *Vandierendonck, Liefooghe & Verbruggen, 2010*). Classic task-switching studies usually involve at least two tasks with specific explanations regarding what task to do at what time and which procedures or rules apply to the stimulus. A switch sequence is defined as two consecutive trials for different tasks, and a repeat sequence is defined as two consecutive trials for the same task. Previous studies have found that people respond more slowly and less accurately to switch sequences than repeat sequences, reflecting the switch cost (*Rogers & Monsell, 1995*; *Monsell, 2003*; *Meiran, 2010*). *Rogers & Monsell (1995)* study included two tasks where either the numerical member of a pair of characters was

Corresponding author
Yonghui Wang,
wyonghui@snnu.edu.cn

classified as even/odd, or the letter member was classified as a consonant/vowel. Their results showed a large asymptotic reaction time (RT) cost. Typically, a task set, or a specific configuration of the cognitive system, is needed to perform a task.

In a typical task-cueing paradigm, a trial consists of a task cue, a stimulus, and a response. The task cue activates the task set, the task set is applied to the stimulus to gain a response, and the response is typically activated via an arbitrary (and recently learned) stimulus (i.e., response mapping) (*Grange & Houghton, 2010*). A task set is defined as a collection of control settings or task parameters that program a system to perform processes such as stimulus identification, response selection, or response execution (*Vandierendonck, Liefooghe & Verbruggen, 2010*). Task sets are also assumed to include a representation of a task goal, a set of task-relevant stimuli, a set of possible responses, and a mapping of the stimulus or stimulus categories to the responses (*Koch et al., 2010*). Therefore, a task set includes two aspects: a stimulus set and a response set. The stimulus set represents stimulus-related processes such as stimulus encoding and identification. The response set represents response-related processes such as relevant stimulus–response (S–R) mapping and activation of the correct response category (e.g., left or right) or modality (e.g., finger or foot) (*Rogers & Monsell, 1995*; *Schuch & Koch, 2004*; *Philipp & Koch, 2005*). For example, in a color judgment task, a left key press may mean "red", but in a size judgment task, a left key press may mean "small". Therefore, S–R mapping constitutes the cognitive meaning of a response, which is the link between a stimulus and a response (*Gade & Koch, 2007*). Changing the correspondence of the S–R mapping and the value of the stimulus would affect the S–R complexity. For example, univalent mappings link one dimension of a stimulus with a response key, while bivalent mappings link two dimensions of a stimulus with a response key.

Several previous studies considered the effect of inhibiting the task set during task switching by applying the backward inhibition paradigm (*Mayr & Keele, 2000*; *Moritz, Hübner & Kluwe, 2004*; *Whitmer & Banich, 2007*). Within this paradigm, participants switch between three tasks, and each currently-relevant task is signaled by a task cue. When switching between tasks A, B, and C, an "ABA" sequence means switching back to a task that was recently performed, and a "CBA" sequence means switching back to a task that was not recently performed. These researchers found that the ABA sequence had slower RTs than the CBA sequence, demonstrating an $n-2$ repetition cost. When participants were required to switch from Task A to Task B, residual activation of the recently-performed Task A impaired the successful implementation of Task B (*Gade & Koch, 2005*). In order to overcome this impairment, Task A must be inhibited (*Mayr & Keele, 2000*). When participants are required to return to Task A, this task remains inhibited, and its execution takes longer (*Gade & Koch, 2008*). Thus, task-related stimulus dimensions are targeted by inhibitory processes once these dimensions are actively abandoned (*Koch et al., 2010*). Furthermore, $n-2$ repetition costs have been seen in classification tasks, response modes, language-defined response sets, and cue-target relations. These studies suggest that $n-2$ repetition costs are markers for inhibitory processes across various levels within a task set (*Koch et al., 2010*).

Previous studies have investigated the role of S–R mapping in generating backward inhibition, and they found that backward inhibition depends on the task switch's response selection stage by adapting the paradigm to include a go/no-go manipulation (*Schuch & Koch, 2003*). Additionally, overlap in a response set affects task inhibition (*Gade & Koch, 2007*), suggesting that when competing tasks share response sets (e.g., the same left and right key presses), the competition between tasks increases and triggers task inhibition to resolve this competition. They found overlap in the response set, thereby influenced the occurrence of task inhibition, reflecting a *n*-2 repetition cost. However, in a task-switching study, *Houghton, Pritchard & Grange (2009)* found that complex (spatially incongruent) S–R mapping reduced overall RTs, but did not affect backward inhibition. Therefore, studies on the effect of S–R mapping on backward inhibition and overlap in a response set on task inhibition have delivered inconsistent results. Moreover, S–R mapping is a subprocess of response selection, and the left inferior frontal gyrus was found to be involved in controlled response selection (*Goghari & MacDonald, 2009*). Backward inhibition also involves response selection (*Schuch & Koch, 2003*), and it has been suggested that backward inhibition processes are mediated via networks consisting of extra striate occipital areas, the temporo-parietal junction, and the inferior frontal gyrus (*Zhang et al., 2016*). The overlap of S–R mapping and backward inhibition in the inferior frontal gyrus suggests that they may affect each other while competing for the same cognitive resources. However, few studies have focused on the impact of S–R mapping on backward inhibition, or have fully explained the influence of S–R complexity on task-set inhibition in one experimental setting. Therefore, we manipulated and generated different S–R complexities in order to investigate the implication of task-set inhibition on task switching. S–R complexity refers to the different stimulation and response difficulties. Whether the response mapping was cued or not reflected the response difficulty, and whether the task was univalent or bivalent reflected the stimulation difficulty. Some studies have also tested this manipulation to define S–R complexity (*Iveson, Tanida & Saito, 2016*). Therefore, we manipulated the response mapping and the task values to define four different types of S–R complexities: univalent stimulus with cued response mapping, univalent stimulus with uncued response mapping, bivalent stimulus with cued response mapping, and bivalent stimulus with uncued response mapping. However, it was too difficult for participants to complete the bivalent stimulus with uncued response mapping, which created a floor effect in our pre-experiment. Therefore, we used the three other types to define S–R complexity in a formal experiment. The easy condition had a univalent task value, only one dimension needing to be discriminated (e.g., color), and a cued response mapping, creating the simplest S–R complexity. The moderate condition also had a univalent task value and only one dimension needing to be discriminated (e.g., color), but the response mapping was uncued, making the S–R complexity moderate. The difficult condition had a bivalent task value, two dimensions needing to be discriminated (e.g., color + shape), and the response mapping was cued. This S–R complexity was the most difficult. Participants were instructed to discriminate between the stimuli dimensions and make the corresponding response. In this study, we investigated the influence of S–R complexity on task-set inhibition in task switching. We manipulated both factors in

the same experiment using the backward inhibition paradigm and by controlling the task sequence and S–R complexity. To implement Task A in Trial n, we inhibited the residual activation of Task A in the $n-2$ trial. Therefore, the inhibition was in the ABA sequence. In this study, the $n-2$ repetition cost illustrated the inhibition effect. When the resources needed for a cognitive activity exceeded an individual's total cognitive resources, this led to cognitive overload and affected the efficiency of the individual's cognitive processing (*Chandler & Sweller, 1991*). Because of this, we expected that more difficult S–R complexities would lead to greater switch costs, and that increasing the difficulty of the S–R complexity would weaken the effect of the task-set inhibition.

## MATERIALS AND METHODS

### Participants

Thirty-two undergraduate students($M_{age}$ = 18.83 years, $SD_{age}$ = 0.81 years) from Shaanxi Normal University were compensated to participate in the experiment. All participants reported normal or corrected visual acuity. They did not know the purpose of the experiment in advance. They signed informed consent that were approved by the Committee on Human Research Protection of the School of Psychology of Shaanxi Normal University (Ethical Application Ref: HR 2019-04-007).

We conducted an a priori power analysis using G*Power (*Faul et al., 2009*) in order to determine the sample size before data collection. The sample size was determined by estimating a moderate effect size for our interaction effect ($f$ = 0.2) and using a power analysis that set the alpha level at 0.05 to determine the number of participants needed to achieve a 0.90 power level with a medium-size effect. The power analysis showed that a sample size of approximately 28 individuals was sufficient to achieve a power level of 0.90. Based on this analysis and on previous studies (*Grange & Houghton, 2010*; *Gade & Koch, 2014*), were cruited 32 participants to participate in our experiment.

### Apparatus and stimuli

The stimuli were presented on a 17-in color monitor with a viewing distance of approximately 50 cm. We set the screen resolution at $1{,}280 \times 1{,}024$ pixels and the refresh rate at 60 Hz. The centric fixation subtended $2.7° \times 2.7°$ and the target traffic signs subtended $2.7° \times 2.7°$.The cues varied across the three tasks and were all presented in one outlined square, subtending $2.7° \times 2.7°$.

Before starting the experiment, all participants were orally informed to consider the traffic sign's shape, color, and texture (but not the meaning), and they were also given written instructions. All participants were given the same information and instructions about the experiment. The cues were different for different tasks of varying difficulty. In the easy condition, the cue was a word in Chinese, either 颜色 (color), 形状 (shape), or纹理 (texture), and the two corresponding keys on the keyboard were marked below the Chinese word. The cue was presented in the center of the screen to remind the participants to focus on this dimension of the stimulus during the discrimination task. In the moderate condition, the cue was one word in Chinese, either 颜色 (color), 形状 (shape), or 纹理 (texture). In the difficult condition, the cue was two words in Chinese,
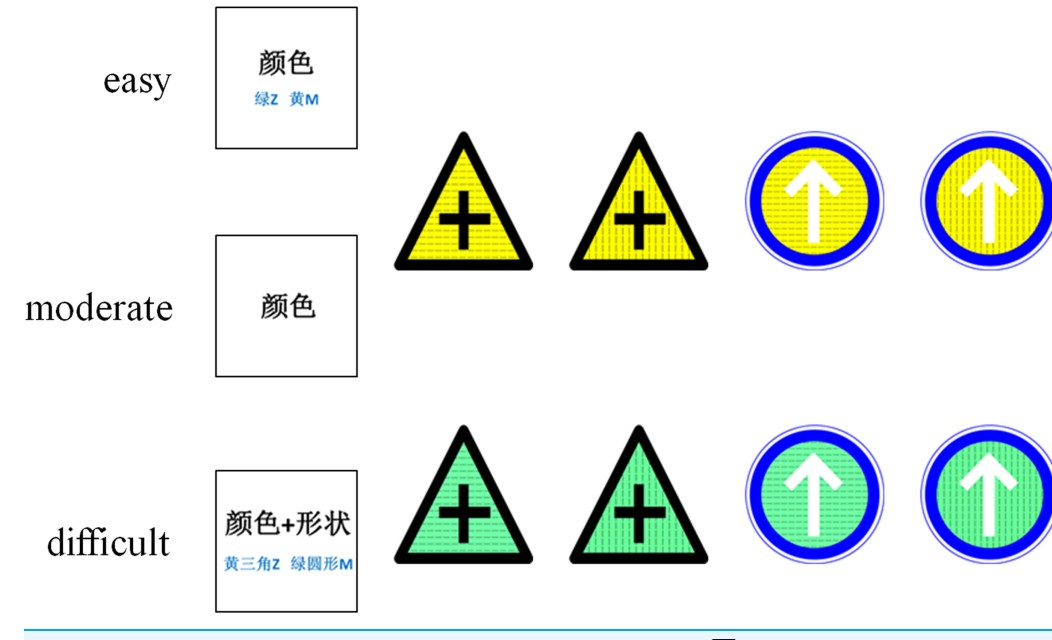

**Figure 1 Stimuli used in experiments.**

either 颜色＋形状 (color + shape), 形状+纹理 (shape + texture), or 纹理+颜色 (texture + color), and two of the four keys corresponding with the dimensions were marked below the Chinese word (Fig. 1). The response keys were based on the conjunctions of the two characters. Participants were tasked to discriminate the stimuli dimension and to make the corresponding response. In the easy and moderate conditions, participants needed to discriminate one stimuli dimension (color, shape, texture) and make their response accordingly. For example, during a color task, the ″m″ key corresponded to the yellow stimuli and the ″z″ key corresponded to the green stimuli. In the difficult condition, participants needed to discriminate two stimuli dimensions (color + shape, color + texture, or shape + texture) and make their response accordingly. For example, the ″m″ key corresponded to the green triangle, yellow horizontal line, and triangle with a vertical line stimuli. The ″z″ key corresponded to the green circle, green vertical line, and circle with a horizontal line stimuli. Two keys were presented on one cue at a time, and all combinations were counterbalanced across participants.

## Design and procedure

The experiment was conducted using E-prime 2.0. We used a3 (task sequence: ABA, CBA, XAA) ×3 (S–R complexity: easy, moderate, difficult) within-subjects factorial design. The labels A, B, and C were used as placeholders, meaning that each task could be repeated. When the sequence had the same task for the $n$ and $n-2$ trials, the sequence was defined as an "ABA switch sequence" (i.e., ABA, ACA, BAB, BCB, CAC, or CBC). When the sequence had different tasks for the $n$, $n-1$, and $n-2$ trials, it was defined as a "CBA switch sequence" (i.e., CBA, BCA, CAB, ACB, BAC, or ABC). Finally, when the sequence had the same task for the $n$ and $n-1$ trials, it was defined as an "XAA repeat sequence" (i.e., ABB, ACC, BAA, BCC, CAA, CBB, AAA, BBB, or CCC).

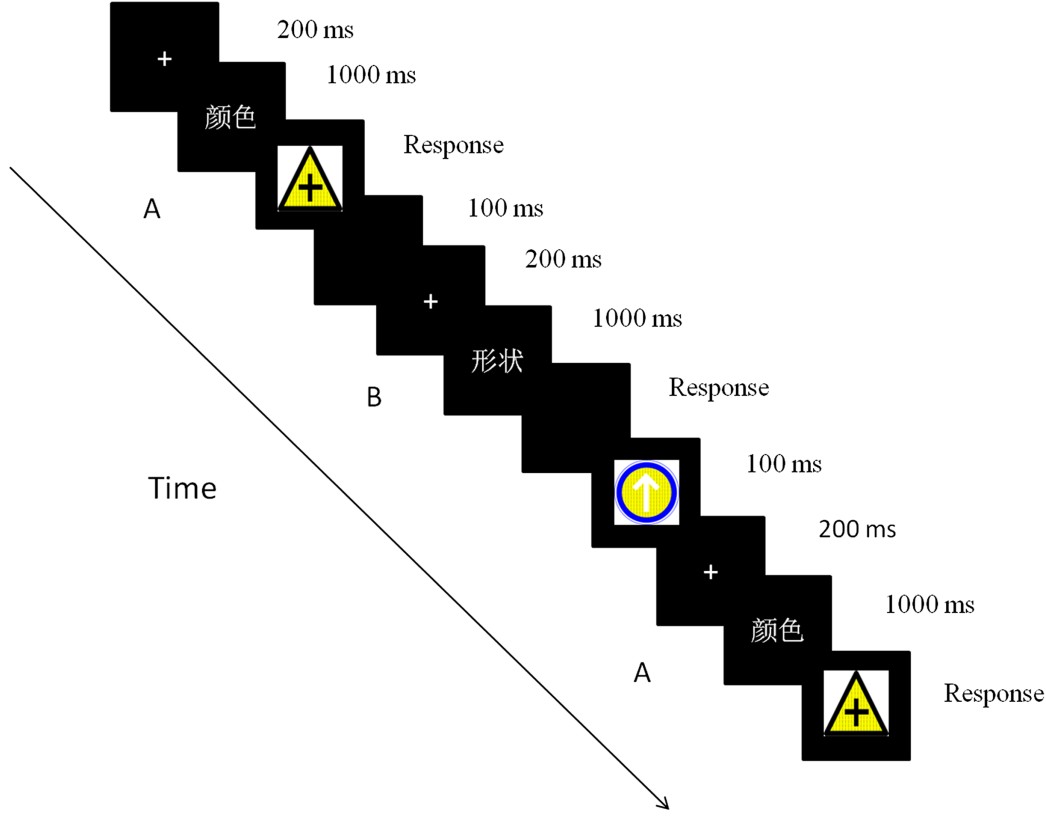

**Figure 2  The time course of a trial in the experiment.**

Each trial started with one fixation presented on a black background for 200ms. The cue then appeared for 1,000 ms to remind the participants to focus on this dimension of the stimuli during the discrimination task. Finally, the target was displayed until the participants responded, and an error response was signaled through feedback. Participants were instructed to respond as quickly as possible while maintaining attention and at least 90% performance accuracy. The next trial started after a blank inter-trial interval of 100 ms. The trials were presented in triplicate, with three trials representing one sequence. The intervals between these triple sequences were 1,500 ms (one-third of the total sequences) or 3,000 ms (two-thirds of the total sequences) (*Dreher & Berman, 2002*). The inter-sequence interval was randomized across the sequences (Fig. 2).

Each participant completed three sessions (easy, moderate, and difficult) for the nine blocks, with 72 sequences per block and three blocks per session. They also completed one 36-sequence practice block per session. Participants were encouraged to take a short break after each block. Given that the difficult task took 50 min to complete, which was almost the time it took to complete both easy and moderate tasks (60 min). Therefore, in order to avoid fatigue, we divided the easy and moderate tasks into one session (session A) and difficult task into another session (session B). And each participant completed two sessions on two separate days, the order of the easy and moderate tasks in session A and the order of the two sessions were counterbalanced across participants.

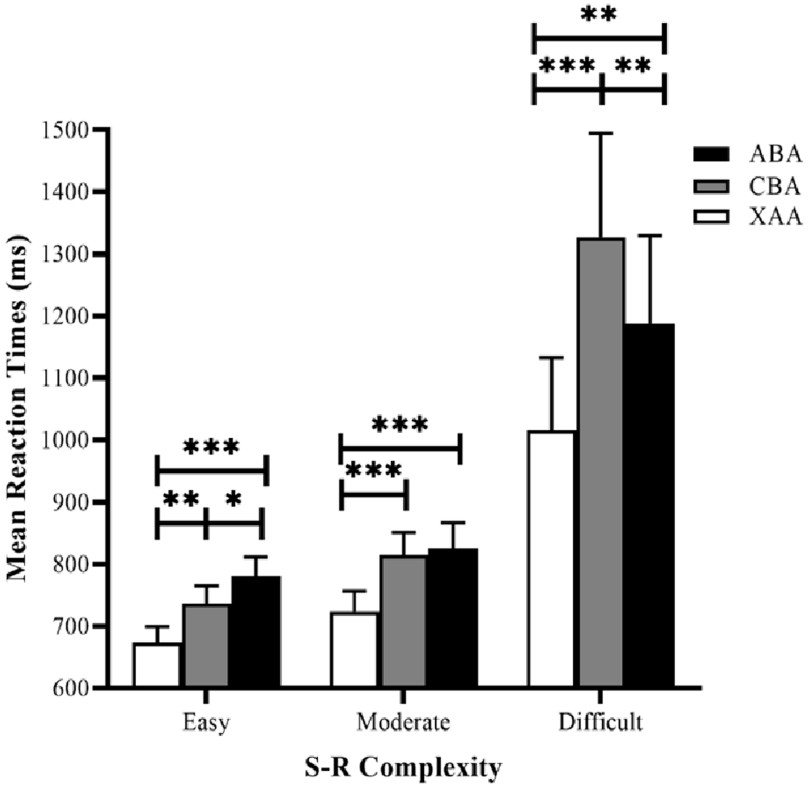

**Figure 3 Mean RTs for correct responses in the experiment as a function of task sequence (ABA, CBA, XAA) and S–R complexity (easy, moderate, difficult). RT, reaction time.** Error bars denote standard errors of the mean. $^*p < 0.05$, $^{**}p < 0.01$, $^{***}p < 0.001$.

## RESULTS

The data were analyzed using SPSS18.0. We only analyzed the RTs for correct responses. The first set of three trials every per block as warming up trials was excluded and RTs with more than three standard deviations were excluded from the analyses; approximately 3.8% of the trials were discarded. All participants performed at levels of >90% accuracy in each condition across all experiments. We used bivariate correlations between the RTs and accuracy data ($r = 0.083$, $p = 0.652$) and found no significant correlations between them, suggesting that the RT results could not be attributed to the tradeoff between response speed and response accuracy.

Reaction time data were submitted to a $3 \times 3$ repeated measures analysis of variance (ANOVA), using task sequence (ABA, CBA, and XAA) and S–R complexity (easy, moderate, and difficult) as the factors (Fig. 3). The results revealed the task sequence's main effect was significant ($F(2, 62) = 27.86$, $p = 0.000$, and $\eta_p^2 = 0.47$). Post hoc tests (LSD) showed that RTs in the XAA sequence were significantly faster than those in the ABA sequence ($p < 0.001$) and the CBA sequence ($p < 0.001$), generating a switch cost. The main effect of S–R complexity was also significant ($F(2, 62) = 9.41$, $p < 0.001$, $\eta_p^2 = 0.23$). Post hoc tests showed that RTs in the easy condition were significantly faster than those in the moderate ($p = 0.013$) and difficult ($p = 0.003$) conditions. RTs in the moderate

**Table 1 Mean error rates (percentage figures) as a function of task sequence and S–R complexity (M ± SD).**

| Task sequence | S–R complexity | | |
| --- | --- | --- | --- |
| | Easy | Moderate | Difficult |
| ABA | 1.07 ± 1.10 | 0.78 ± 0.96 | 1.35 ± 1.12 |
| CBA | 1.07 ± 1.34 | 1.20 ± 1.28 | 1.95 ± 1.73 |
| XAA | 1.17 ± 1.54 | 0.94 ± 0.96 | 1.39 ± 1.32 |

condition were significantly faster than those in the difficult condition ($p = 0.007$), suggesting a valid S–R complexity. More importantly, the interaction between task sequence and S–R complexity was significant ($F (4, 124) = 9.29$, $p < 0.001$, $\eta_p^2 = 0.23$). Simple effect analyses showed that in the easy condition, the effect of the task sequence was significant ($F (2,62) = 16.90$, $p < 0.001$). Post hoc tests with LSD showed RTs in the XAA sequence were significantly faster than those in the ABA sequence ($p < 0.001$)and the CBA sequence ($p = 0.001$), yielding a switch cost. Moreover, RTs in the CBA sequence were significantly faster than those in the ABA sequence ($p = 0.045$), reflecting an $n-2$ repetition cost. In the moderate condition, the effect of the task sequence was significant ($F (2, 62) = 15.12$, $p < 0.001$), and post hoc tests showed that RTs in the XAA sequence were significantly faster than those in the ABA sequence ($p < 0.001$) and the CBA sequence ($p < 0.001$), yielding a switch cost. However, the RTs did not significantly differ between the CBA and ABA sequences ($p = 0.632$). In the difficult condition, the effect of the task sequence was significant ($F (2, 62) = 17.12$, $p < 0.001$), and post hoc tests showed that RTs in the XAA sequence were significantly faster than those in the ABA sequence ($p = 0.003$) and the CBA sequence ($p < 0.001$), yielding a switch cost. Moreover, RTs in the ABA sequence were significantly faster than those in the CBA sequence ($p = 0.003$), reflecting an $n-2$ repetition facilitation.

Error rate data (Table 1) were also submitted to a 3 × 3 repeated measures ANOVA, using task sequence (ABA, CBA, XAA) and S–R complexity (easy, moderate, difficult) as the factors. The main effect of the task sequence was significant ($F (2, 62) = 6.15$, $p = 0.004$, $\eta_p^2 = 0.17$). Post hoc tests showed that error rates in the CBA sequence were significantly greater than those in the ABA sequence ($p = 0.003$) and the XAA sequence ($p = 0.032$). The main effect of S–R complexity was also significant ($F (2, 62) = 5.43$, $p = 0.007$, $\eta_p^2 = 0.15$). Post hoc tests showed that error rates in the difficult condition were significantly greater than those in the moderate ($p = 0.001$) and easy ($p = 0.045$) conditions. The two-way interaction between task sequence and S–R complexity was not significant ($F (4, 124) = 2.41$, $p = 0.053$, $\eta_p^2 = 0.07$). Simple effect analyses showed that in the easy condition, the effect of the task sequence was not significant ($F (2, 62) = 0.25$, $p = 0.778$). In the moderate condition, the effect of task sequence was significant ($F (2, 62) = 3.58$, $p = 0.034$), and post hoc tests showed that error rates in the CBA sequence were greater than those in the ABA sequence ($p = 0.001$). In the difficult condition, the effect of the task sequence was significant ($F (2, 62) = 6.32$, $p = 0.003$), and post hoc tests showed that

error rates in the CBA sequence were larger than those in the ABA sequence ($p = 0.012$) and the XAA sequence ($p < 0.004$).

To verify the stability of the $n-2$ repetition facilitation in the difficult condition, we recruited another 32 participants ($M_{age}$ = 19.53 years, $SD_{age}$ = 0.83 years) from Shaanxi Normal University to conduct an additional experiment with the difficult condition. Only RTs for correct responses were analyzed and RTs of more than three standard deviations were excluded from the analyses; thus, approximately 4.1% of the trials were discarded. RTs and error rate were submitted to a one-way ANOVAs with task sequence (ABA, CBA, XAA) as factor. The ANOVAs conducted on RTs showed a significant main effect of the task sequence ($F$ (2, 62) = 7.02, $p = 0.002$, $\eta_p^2 = 0.19$). Post hoc tests with LSD showed that RTs in the XAA sequence ($M$ = 936.47 ms, SE = 60.88) were significantly faster than those in the CBA sequence ($M$ = 1012.65 ms, SE = 75.86) ($p = 0.005$), yielding a switch cost, and the ABA sequence ($M$ = 953.76ms, SE = 63.33) were significantly faster than those in the CBA sequence ($M$ = 1012.65 ms, SE = 75.86) ($p = 0.02$), reflecting an $n-2$ repetition facilitation. The ANOVAs conducted on error rate showed that the main effect of the task sequence was not significant ($F$ (2, 62) = 0.41, $p = 0.66$, $\eta_p^2 = 0.01$). Therefore, the results of this additional experiment replicated an $n$-2 repetition facilitation effect in the difficult condition..

## DISCUSSION

Our study manipulated S–R complexity and task sequence types in order to investigate the influence of S–R complexity on task-set inhibition in task switching. The RTs were faster in the repeated sequence than in the switch sequence. More importantly, the switch costs differed significantly across the three S–R complexity levels. Specifically, the switch cost was greater in the difficult condition than in the moderate and easy conditions.

Stimulus–response complexity affected the switch cost. Researchers have suggested that switch costs only occur when the underlying S–R mappings are represented as task sets (*Dreisbach, Goschke & Haider, 2007*). In our study, participants were tasked to remember the S–R mappings, strengthen this memory through practice, and then construct the corresponding task set following the instructions. When the cues were presented, participants needed to retrieve the task set, discriminate the dimensions of the stimulus, and perform a task according to the S–R mappings. In the difficult condition, the S–R complexity was more difficult because the S–R mapping associated two dimensions of the task with a corresponding response key, meaning that the participants needed to correctly select two stimulus characters and one of four response keys. However, S–R mapping was easier in the easy and moderate conditions because it associated one dimension of the task with a corresponding response key, meaning that only one stimulus character and one of two response keys needed to be discriminated. Therefore, both the RTs and switch costs were greater in the difficult condition.

In our study, the influence of S–R complexity on task-set inhibition demonstrated different modes via different S–R complexity levels. When two components share common

brain regions in the inferior frontal gyrus, they compete for cognitive resources and each component is assigned different resources. The easy condition had a significant $n-2$ repetition cost. According to the account of persisting inhibition, when the $n-2$ trial (Task A) was completed, Task A remained activated. To overcome the residual activation of Task A, subjects needed to suppress it when required to return to Task A (Trial $n$). With easier S–R complexity, the participants allocated fewer cognitive resources to discriminate one dimension of the stimulus and select the corresponding keys. Therefore, the brain resources for inhibiting the $n-2$ trial task were sufficient. However, the task was still inhibited and took longer to implement, creating slower RTs for the ABA sequence. In the difficult condition, the $n-2$ repetition cost was reversed, similar to an $n-2$ repetition benefit, suggesting that residual activation from a previously performed task (i.e., trial $n-2$, Task A) lessened the accumulating information for the same task in the n trial. When the $n-2$ trial task reappeared in the n trial, participants allocated more cognitive resources to handle the S–R complexity. For example, they would discriminate between two dimensions of the stimulus and select the corresponding keys. Therefore, the cognitive resources for inhibiting the $n-2$ trial task were insufficient, causing residual activation to persist. When this $n$ trial task reappeared, a facilitated effect was obtained. These results were consistent with those of previous studies, suggesting that distracted inhibition is stronger with an increased cognitive load (*Conway et al., 1999*; *Gibbons & Stahl, 2010*).

This study had some limitations. First, inhibited ability, a high-level cognitive function, is closely related to the prefrontal lobe, which is not fully developed until adulthood (*Luna, Padmanabhan & O'Hearn, 2010*). Our participants were college students who reached adulthood, but children would not have fully developed inhibited abilities compared to the receded inhibited abilities of older participants. Therefore, future studies should investigate the influence of S–R complexity on task-set inhibition across different age groups from the developmental perspective. Second, the prefrontal cortex, anterior cingulated cortex, pre-supplementary motor area, and posterior parietal cortex were activated during tasks witching (*Dove et al., 2000*; *Duncan, 2001*; *Johnston et al., 2007*; *Shi et al., 2010*), and the prefrontal lobe was involved in inhibition and S–R mapping. The brain mechanisms of inhibition across different S–R complexity levels require further study.

## CONCLUSION

This study investigated the impact of S–R complexity on task-set inhibition in task switching. The switch costs under difficult conditions were greater than those under moderate and easy conditions, suggesting that S–R complexity affects task switching. Specifically, S–R complexity affected task-set inhibition. The easy condition showed a significant $n-2$ repetition cost, while the difficult condition showed an $n-2$ repetition benefit, suggesting that an increased S–R complexity does not make the task-set inhibition significant. We suggest that S–R complexity and inhibition compete for cognitive resources, and that these different resource assignments led to our results.

### Funding

This work was supported by the Fundamental Research Funds for the Central Universities (No. 2019TS136) and the National Natural Science Foundation of China (Nos. 31700945 and 31800910). The funders had no role in study design, data collection and analysis, decision to publish, or preparation of the manuscript.

### Grant Disclosures

The following grant information was disclosed by the authors:
Central Universities: 2019TS136.
National Natural Science Foundation of China: 31700945 and 31800910.

### Competing Interests

The authors declare that they have no competing interests.

### Author Contributions

- Li Zhao conceived and designed the experiments, performed the experiments, analyzed the data, prepared figures and/or tables, authored or reviewed drafts of the paper, and approved the final draft.
- Saisai Hu conceived and designed the experiments, analyzed the data, prepared figures and/or tables, authored or reviewed drafts of the paper, and approved the final draft.
- Yingying Xia performed the experiments, prepared figures and/or tables, and approved the final draft.
- Jinyu Li performed the experiments, prepared figures and/or tables, and approved the final draft.
- Jingjing Zhao analyzed the data, authored or reviewed drafts of the paper, and approved the final draft.
- Ya Li analyzed the data, prepared figures and/or tables, and approved the final draft.
- Yonghui Wang conceived and designed the experiments, authored or reviewed drafts of the paper, and approved the final draft.

### Human Ethics

The following information was supplied relating to ethical approvals (i.e., approving body and any reference numbers):

The Committee on Human Research Protection at the School of Psychology of Shaanxi Normal University granted ethical approval to carry out the study within its facilities (Ethical Application Ref: HR 2019-04-007).

### Data Availability

Raw measurements are available in the Supplemental Files.

## Supplemental Information

Supplemental information for this article can be found online at http://dx.doi.org/10.7717/peerj.10988#supplemental-information.

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
