# Peer review of "Stimulus–response complexity influences task-set inhibition in task switching"

_PeerJ, doi:10.7717/peerj.10988_

## Round 0.1 · original submission · Major Revisions

The reviewers particularly stressed the need to better explain some methodological details and to check how the statistics were used. Authors are also advised to better control the use of English which has been found to be inadequate.

·

Basic reporting

Although I could follow most of the manuscript, I thought the English language could be improved. The manuscript is, at some points, difficult to follow, and this would probably even be more the case for readers unfamiliar with the topic.

Experimental design

The present manuscript tested the influence of stimulus-response complexity on the backward inhibition effect. While I found the general idea behind this study interesting, I am afraid I found it difficult to evaluate, because I do not completely understand the task that participants had to perform, and how stimulus-response complexity was manipulated. Specifically, my main comment is the following:

The complexity manipulation was not completely clear to me. S-R complexity can be manipulated in various ways, so I believe it is important to be clear on this. The manipulation is introduced in the introduction, but it is unclear to me how a task can be mapped to only one response key (lines 101-105)? If I understand the tasks correctly, complexity was also manipulated in a different way when comparing easy to medium conditions (i.e., cued versus uncued response mappings), versus medium to difficult conditions (i.e., univalent versus bivalent stimuli/tasks). Does this also mean that the tasks were the same for the easy and medium condition, but not the difficulty condition? Most importantly, I am unsure about what exactly the participants were instructed to do.

Minor:

On lines 136-138: Can the authors clarify what is meant with “or informed also in the instruction”? Does this mean that some participants were only informed about some of the information on screen, and others not? If so, was this an important manipulation?

On lines 175-176, can the authors be more explicit on which orders were possible and which not. I now understand only E(asy)-M(edium)-D(ifficult) and D-M-E were possible? Or were M-E-D and D-E-M possible, too?

I found the label “CAA repeat sequences” a bit misleading, if these can also include AAA (or BBB or CCC) sequences. Maybe it would be best to label them XAA sequences or AA sequences?

This is a very minor comment, but I believe it is technically incorrect to state that a p-value = .000. It can only be smaller than .001.

If I understand the introduction correctly, the study by Gade & Koch (2007) would have hypothesized to find the opposite effect (or even found the opposite effect themselves?). However, this discrepancy is not really being discussed.

Signed,

Senne Braem

Validity of the findings

Minor:

I did not find the argument that “all subjects having an accuracy above 90% should do away with speed-accuracy trade-off concerns” very convincing. Along those lines, I believe it would be good to briefly unpack the direction of the marginally significant two-way interaction in the error rates.

The sample size seems justified, but a replication would be reassuring, especially if these findings do indeed go against what would be hypothesized based on Gade & Koch (2007, see below). Along those lines, I would be interested in knowing how much this effect might be dependent on the broader context that is being created by manipulating the different complexity conditions within subjects: This could be tested by only testing the difficulty condition in a separate sample and again show the reversed backward inhibition effect.

Reviewer 2 ·

Basic reporting

In the work “Stimulus-response complexity influences task-set inhibition in task switching” Zhao et al. asked how the complexity S-R association interact with the backward inhibition cost in a task switching task. The authors observed that the reaction time was mainly influenced by both the task sequence and the S-R complexity. In addition, the authors detected a significant interaction between these two variables revealing that the n-2 switching observed in the two easier S-R condition (easy and moderate S-R complexity) was canceled in the difficult condition. The authors discuss their results in line with the hypothesis that different cognitive resources and neural circuits are recruited depending on the different degrees of complexity of the task.
I found the paper well introduced and referenced and the experimental design appropriated to the study. However there are several missing information in the methods that I would the authors to provide, before taking a decision on the reliability of their results.

Experimental design

The main issue of the present work is the lack of information on the statistical analysis applied. More specifically, the information on the post-hoc comparisons is not indicated in the method section and it is only reported in the results section. There the authors report different methods for testing the difference between conditions.
In particular, when the authors perform the post-hoc comparisons for studying the main effects of task sequence and S_R complexity they declare to apply a Fisher’s LSD test which is a test adjusted for multiple comparisons. However, when the authors studied the interaction effect, the authors applied multiple t-test comparisons not corrected for multiple comparisons. In the latter case the authors are exposed to Type I errors accepting a false positive in place of a significant effect not present in the data. If this is the case, their results might be differently interpreted.
My suggestion is to repeat each post-hoc evaluation by applying the same multiple comparisons method, i.e. use the Fisher’s LSD comparison for all of them.
As minor point, the authors should refer to the software used to run the experimental procedure an to perform the statistical tests.
The information reported in the result section are redundant. The mean and standard error of the same group is reported several times and this repetition is confusing. In addition, the authors report in the text the same values reported in Figure 3. My suggestion is to report in the results section only the results of the statistical tests and refer to Figure 3 to get information on mean and variability of each condition. Related to this, the addition of graphical markers to Figure 3 for highlighting significant conditions’ different will help the reader in interpreting the results.

Validity of the findings

The validity of the finding should be reconsidered in light of the new analysis (see above) and in case discuss a negative result.

---

## Round 0.2 · Minor Revisions

Please refer to the requirements of Reviewer 1.

·

Basic reporting

The reporting is much improved, the article is much clearer to me now. Thanks!

Experimental design

I have no further suggestions or comments on the design.

Validity of the findings

I believe the results are clearly presented, and I would like to thank the authors for providing a replication of the effect in the difficulty condition. I would suggest to also mention this replication in the abstract, as well as providing the details of the participants in the participant section and the other results of that experiment in the results section (e.g., average accuracy, error rates).

Additional comments

Thank you for considering my comments and suggestions. I believe the manuscript is definitely improved. I only have two remaining minor comments:

This newly added sentence is confusing to me: "The order of the experiment included four types: Easy (E)-Medium (M)-Difficult (D), D-M-E, M-E-D, and D-E-M. The difficult task took 50 min to complete, which was almost the time it took to complete both easy and moderate tasks (60 min). Therefore, in order to avoid fatigue, we divided the participants into four groups that each completed one type of task.".
It now sounds as if participants only performed one of the different difficulty conditions (i.e., "one type of task"). Also, I am not sure what is meant with "in order to avoid fatigue". Is this to suggest that the alternative could have been that all participants had to go through all four possible task orders? Or does the fatigue-argument refer to why the M-D-E or E-D-M order was not used? I would suggest revising these two sentences to make this more clear.

Second, the newly added experiment could be detailed a bit more. I would propose mentioning this replication of the reversed backward inhibition effect in the abstract, discussing its results a bit more extensively, and to provide some further details in the method/participant section.

Reviewer 2 ·

Basic reporting

The authors addressed all my questions.

Experimental design

My questions about experimental design and data analysis have been answered.

Validity of the findings

The new analysis supports the authors' discussion.

Additional comments

I do not have further comments.

---

## Round 0.3 · accepted · Accept

Please consider the suggestions, included in the last comments, received by Reviewer 1 and submit the final version.

·

Basic reporting

no comment

Experimental design

no comment

Validity of the findings

no comment

Additional comments

Thank you for making these changes.

I hope to be of help with these two final spelling suggestions:

"We also replicate a reversed n-2 repetition cost."
->
"These results replicated the reversed n-2 repetition cost."

"The results of the additional experiment replicated an n−2 repetition facilitation in difficult condition."
->
"Therefore, the results of this additional experiment replicated the n−2 repetition facilitation effect in the difficult condition."